# Thimerosal-Preserved Hepatitis B Vaccine and Hyperkinetic Syndrome of Childhood

**DOI:** 10.3390/brainsci6010009

**Published:** 2016-03-15

**Authors:** David A. Geier, Janet K. Kern, Brian S. Hooker, Lisa K. Sykes, Mark R. Geier

**Affiliations:** 1Institute of Chronic Illnesses, Inc., Silver Spring, MD 20905, USA; davidallengeier@comcast.net (D.A.G.); mgeier@comcast.net (M.R.G.); 2CoMeD, Inc., Silver Spring, MD 20905, USA; syklone5@verizon.net; 3Council for Nutritional and Environmental Medicine, Mo i Rana, Norway; 4Department of Biology, Simpson University, Redding, CA 96003, USA; bhooker@simpsonu.edu; 5Focus for Health, Watchung, NJ 07069, USA

**Keywords:** ADHD, Thimerosal, neurodevelopmental disorder, Hyperkinetic syndrome of childhood

## Abstract

(1) Background: Hyperkinetic syndrome of childhood (HKSoC) is an International Classification of Diseases, Ninth Revision, Clinical Modification (ICD-9) category in which the majority of the children are also diagnosed under the Diagnostic and Statistical Manual of Mental Disorders, 4th Edition, Text Revision (DSM-IV-TR), where the umbrella term is “Attention-Deficit and Disruptive Behavior Disorders”. The diagnostic criteria for HKSoC are developmentally inappropriate inattention, hyperactivity, and impulsivity. Some studies have implicated mercury (Hg) exposure as a risk factor. (2) Methods: This hypothesis testing study; using the Vaccine Safety Datalink; assessed the toxicological effects of bolus exposure to organic-Hg from Thimerosal-containing vaccines (TCVs) by examining the relationship between Thimerosal-preserved hepatitis B vaccines (TM-HepB) given at varying levels and at specific intervals in the first six months after birth and the risk of a child being diagnosed with HKSoC. (3) Results: Children diagnosed with HKSoC were significantly more likely to be exposed to increased organic-Hg from TM-HepB doses given within the first month (odds ratio = 1.45; 95% confidence interval (CI) = 1.30–1.62); within the first two months (odds ratio = 1.43; 95% CI = 1.28–1.59); and within the first six months (odds ratio = 4.51; 95% CI = 3.04–6.71) than controls. (4) Conclusion: The results indicate that increasing organic-Hg exposure from TCVs heightens the risk of a HKSoC diagnosis.

## 1. Introduction

Hyperkinetic syndrome of childhood (HKSoC) is behaviorally defined in the International Classification of Diseases, Ninth Revision, Clinical Modification (ICD-9) [1]. HKSoC is an ICD-9 category (314.xx) for the group of conditions, which include the following six subtypes: attention deficit disorder (ADD) without mention of hyperactivity (314.00); attention deficit disorder with hyperactivity (ADHD) (314.01); hyperkinesis with developmental delay (314.1); hyperkinetic conduct disorder (314.2); other specified manifestations of hyperkinetic syndrome (314.8); and unspecified hyperkinetic syndrome (314.9). The specific diagnostic criteria are features of developmentally inappropriate inattention, hyperactivity, and impulsivity [2]. The disorder is characterized by a marked pattern of inattention and/or hyperactivity-impulsivity that is inconsistent with developmental level and clearly interferes with normal functioning in at least two settings (e.g., at home and at school). At least some of the symptoms must be present before the age of seven years. Most children in this category are labeled with the Diagnostic and Statistical Manual of Mental Disorders, 4th Edition, Text Revision (DSM-IV-TR) diagnostic term ADHD and, to a lesser extent, ADD [2]. Moreover, to date, most of the published studies have been conducted on those who have an ADHD diagnosis.

Although these are the features used to define HKSoC and its major subcategory ADHD, many studies also report difficulties in social interaction and communication and language [3,4]. These children are more likely to have problems in school, have strained family and peer-relationships, and suffer more injuries than children without the disorder [4,5]. It has been estimated that boys are two to three times more likely to have the disorder than girls [6].

According to the American Academy of Pediatrics (AAP), which tends to use the DSM-IV terminology, ADHD is a chronic condition, and a large proportion of these children continue to meet diagnostic criteria as adults [7]. As of 2007, 66.3% of children with a current diagnosis of ADHD were receiving various medications as a treatment for this chronic condition [5].

Thus, ADHD is one of the most prevalent of all psychiatric disorders [8]. A worldwide prevalence estimate is 5.29% [9]. The United States Centers for Disease Control and Prevention (CDC) states that the percentage of children estimated to have ADHD has increased over time [5]. For example, a 2003 survey of parents (the National Survey of Children's Health) found an estimated 7.8% of children in the US aged 4–17 years had been given a psychiatric diagnosis in this category, which had increased to approximately 9.5% of children 4–17 years of age with a diagnosis in the ADHD category as of 2007 [5].

In fact, there has been a dramatic rise in neurodevelopmental disorders in general in the past two decades driven by the marked increase in both ADHD and autism spectrum disorder (ASD) [10]. It was reported that, from 1997 through 2008, the percent change in the prevalence for ADHD was 33% and 289% for ASD [10]. These two categories of neurodevelopmental disorders have become increasingly prevalent, with high personal, familial, and societal costs. [11]. In a recent study by Grzadzinski *et al.* [4], where they found similar symptomatology in children diagnosed with ASD or ADHD, the authors questioned whether ASD extends into ADHD. It has been suggested that these two psychiatrically derived conditions fall on the same continuum [4,12].

There is, at present, no agreement on the cause of HKSoC. Because it sometimes runs in families, genetics is apparently a factor, but there are also epigenetic (environmental) factors involved. Some studies have implicated toxic metal exposure, predominately lead (Pb) and mercury (Hg), as a risk factor. Low-level Pb exposure has been associated with a clinical diagnosis of ADHD in several studies [13]. For example, ADHD symptoms are found to increase with blood-Pb concentration [14,15].

Although there are general Hg-exposure studies implicating Hg as a risk factor in ADHD [16], studies have also shown that Hg exposure from Thimerosal (TM) was a risk factor for neurodevelopmental disorders including ADHD (TM is a preservative used in some vaccinations that is about 49.55% Hg by weight) [17,18,19]. However, to date, this issue remains under debate [20].

This hypothesis testing study further assessed the toxicological effects of bolus exposure to organic-Hg from Thimerosal-containing vaccines (TCVs) using the database from the Vaccine Safety Datalink (VSD). Specifically, the study examined the relationship between Thimerosal-preserved hepatitis B vaccines (TM-HepB) that were given at specific intervals within the first six months after birth and the risk of a child eventually being diagnosed with HKSoC.

## 2. Methods

### 2.1. The Database from the Vaccine Safety Datalink and Study Approval

The National Center for Health Statistics Research Data Center in Hyattsville, Maryland, United States of America (USA), was the secure site where the data were analyzed, in October of 2013. Approval was obtained from the Institutional Review Boards of Kaiser Permanente (KP) Northwest and Northern California. In addition, the study received approval from the CDC. CDC and KP do not necessarily agree with the statements in this study. This database is available to outside researchers after obtaining approval from the CDC and KP. Information regarding the VSD database and access to it is available online [21].

### 2.2. How the Population at Risk Was Determined

Utilizing SAS^®^ software (Statistical Analysis Software (SAS), Version 9.1, SAS Institute, Cary, North Caroline, NC, USA) and the VSD database [21], this study examined more than one million children enrolled at KP Colorado, Northwest, or Northern California (updated through 2000). The VSD database is from the VSD project that began in 1991. This project was a joint endeavor by the CDC and the National Immunization Program. Linking computerized databases of several managed care organizations (MCOs), the VSD provides certain demographic information, detailed vaccination records, and other medical information. The cohort used in this study had to be continuously enrolled from birth in one of these MCOs and had to have date of birth and gender information on record in the system. No other exclusion criteria were used.

### 2.3. How Cases Were Determined

To locate the initial cases of a diagnosis that fell within the HKSoC spectrum (ICD-9 code: 314.xx), including the following subtypes: attention deficit disorder without mention of hyperactivity (314.00), psychiatrically known as ADD; attention deficit disorder with hyperactivity (314.01), psychiatrically known as ADHD; hyperkinesis with developmental delay (314.1); hyperkinetic conduct disorder (314.2), and other specified manifestations of hyperkinetic syndrome; (314.8) and unspecified hyperkinetic syndrome (314.9), the outcome files were examined. This included both outpatient and inpatient diagnoses. When multiple cases of HKSoC umbrella in a child were discovered, only the initial one was used. Table 1 summarizes the year of birth of the children diagnosed with HKSoC identified in the present study. Among those children diagnosed with an HKSoC identified in Table 1, only children where the HKSoC diagnosis came after they received a HepB vaccine were allowed in the analyses. This step was incorporated to be sure of the necessary temporal cause and effect relationship.

A total of 1485 children were identified who had been diagnosed with HKSoC (males = 1158, females = 327, male/female ratio = 3.54:1). Their year of birth ranged from 1991 to 2000. These children who had been given a diagnosis of HKSoC were initially assessed to find the mean age of their initial HKSoC diagnosis (5.72 years of age) and also the standard deviation (SD) of the mean age of their initial HKSoC diagnosis (1.83 years).

### 2.4. How Controls Were Determined

To find control children who did not have an HKSoC diagnosis and only a low probability of getting that diagnosis later as they were followed-up, the control children had to be enrolled continuously from after birth up until they were at least 7.55 years old (mean age of initial HKSoC diagnosis + SD of mean age of initial HKSoC diagnosis). When this rule was applied, it left a group of control children numbering 20,584, with males = 10,303, females = 10,281, and male/female ratio = 1.002. Their year of birth ranged from 1991 to 1993. Thus, the exclusion criteria for the control children (those without a diagnosis of HKSoC) were lack of continuous enrollment, lack of record of the child’s gender, and an age of less than 7.55 years. The year of birth of the control children used in the analyses are summarized in Table 1.

### 2.5. Exposure to HepB Vaccine

For cases and controls, the vaccine file was checked for the exact dates for the administration of the HepB vaccine. The study allowed for children who were not given any HepB vaccine to be included. For cases and controls, the Hg exposure dose was assigned as 12.5 µg Hg for children who were given pediatric TM-HepB. The Hg exposure dose was assigned as 0 µg Hg for children who were given combined *Haemophilus influenzae* type b (Hib)-hepatitis B vaccine (a no-TM hepatitis B vaccine (no-TM-HepB)). Finally, the Hg exposure dose was assigned as 0 µg Hg for children who were not given either of the vaccine types just mentioned.

### 2.6. Statistical Analyses

For the statistical analyses, the SAS^®^ software Fisher's exact test was used. A two-sided *p*-value < 0.05 was assigned as being statistically significant. The null hypothesis for each analysis was that there would be no differences in Hg exposure among HKSoC cases and controls.

In Analysis I, the frequency of exposure 12.5 µg Hg from a TM-HepB within the first month after birth was compared to the frequency of exposure to 0 µg Hg within the first month of life among cases and controls was examined. In Analysis II, the frequency of exposure to 25 µg Hg from two doses of TM-HepB was compared to the frequency of exposure to 0 µg Hg within the first two months of life for cases and controls. In Analysis III, the frequency of exposure to 37.5 µg Hg from three doses of TM-HepB was compared to the frequency 0 µg Hg within the first six months of life for cases and controls. The same analyses were repeated by separately analyzing the data among males (Analysis IV–VI) and females (Analysis VII–IX).

## 3. Results

Table 2 reveals the association between Hg exposure at several specific points within the first six months after birth among cases and controls. The results from Analysis I indicated that cases with an HKSoC diagnosis were significantly more probable (odds ratio = 1.45, *p* < 0.001) than controls to have received 12.5 μg than 0 μg Hg within the first month after birth. Similarly, the outcomes for Analysis II found that cases with an HKSoC diagnosis were significantly more probable (odds ratio = 1.43, *p* < 0.001) than controls to have received 25 μg Hg than 0 μg Hg within the first two months after birth. Lastly, Analysis III found that cases diagnosed with an HKSoC diagnosis were much more likely (odds ratio = 4.51, *p* < 0.001) than controls to have received 37.5 μg Hg than 0 μg Hg within the first six months after birth.

Table 3 and Table 4 reveal the association between Hg exposure at several specific points within the first six months after birth among cases and controls separated by gender. Overall, it was observed that similar associations were observed for between increasing Hg exposure at several specific points within the first six months after birth among cases and controls when the data were separated by gender.

## 4. Discussion

The United States Food and Drug Administration and CDC previously stated that vaccinations should be held to high safety standards [22,23,24]. Because of this, the VSD project was developed to facilitate research on the risks associated with vaccines (hypothesis testing). This current hypothesis testing, comparative case-control study assessed the possible association between a diagnosis of HKSoC and increasing doses of organic-Hg exposure from TCVs using the prospectively collected medical records of subjects within the VSD.

This study exploited the timing of vaccine administration that varied widely as a means to evaluate the potential adverse consequences of Hg exposure from TCVs. The specific intervals examined during the infant period, as well as differences in the collective doses of organic-Hg received, were the consequence of the recommended use of HepB vaccine in infancy [25]. The Advisory Committee on Immunization Practices (ACIP), in 1991, decided that for HepB, the first dose should be given at birth to two months of age, the second dose should be given at one to four months of age, and third dose should be given at six to eighteen months of age [25]. As a consequence, it is worth noting that the variability in the exposure to organic-Hg in all of the analyses in this study was, at least in part, based on these varying and overlapping windows recommended for HepB vaccines.

Furthermore, the ages of vaccine administration studied were selected on the premise that earlier increasing Hg exposure due to TM-HepB vaccinations would be associated with increasing risks for adverse neurodevelopmental outcomes. The study results seem to bear out this premise, and are consistent with previous studies suggesting that there are particularly vulnerable brain developmental periods within the first 6 months of life [26].

The biologically plausible role of organic-Hg exposure from TM and the pathogenesis of a diagnosis of HKSoC are supported by numerous recent studies [27]. Some evidence is drawn from animals and infants studies that examine the distribution and form of Hg from TM after administration.

For example, it was revealed in a study that examined the distribution of Hg from TM by giving TM to infant monkeys based on the vaccination schedule for children in the 1990s that significant levels of Hg were present in the brain and persisted at significant levels more than 120 days after the last exposure [28]. Similar finding were reported in rats [29]. It was even observed in the rats, although TM is essentially ethyl-Hg, that a significant portion of Hg in the brain of the rats was in the methyl-Hg form (there is general consensus that methyl-Hg is very poisonous) [29]. It is also worth noting that ethyl-Hg is actively transported the same as methyl-Hg across neuronal cellular membranes by the L-type neutral amino acid carrier transport (LAT) system [30].

In human infants, TM exposure from vaccines significantly increases the infant’s blood and hair Hg levels. Moreover, some infants vaccinated with TCVs have been found to have blood [31,32,33] and hair [34] Hg levels that were higher than the safety limit set by United States Environmental Protection Agency (EPA).

Hg causes neuronal axons to degenerate because Hg disrupts the structure of the axon, causing it to break apart and depolymerize [35,36,37,38]. As a consequence, the axons degenerate. Furthermore, the effect that Hg has on microtubules and the subsequent axonal degeneration is unique to very low-concentrations of Hg. Other toxic metals at very low concentrations, e.g., Pb, manganese (Mn), cadmium (Cd), and aluminum (Al), do not show this effect [38].

Moreover, studies show that large myelinated axons are selectively vulnerable to damage. It is important to note that projections or long-range neurons have, in general, bigger cell bodies and axons than local circuit (LC) neurons [39,40]. Evidence also suggests the brain tries to regenerate in the face of the loss of long-range axons with a resulting increase in numbers of small axons [41,42,43]. This shift from long-range connectivity to short-range connectivity has been shown in research. Using electroencephalography (EEG), Barttfeld *et al.* [44], for example, in assessing dynamic brain connectivity focusing in the low-frequency (delta) range, found that as long-range connections decreased, there was an increase in short-range connections.

To this point, many studies show abnormal functional connectivity in ADHD, with a relatively consistent loss of long-range axons and an increase in short-range axons in different areas of the brain [8]. Functionally, loss of long-range axons and an increase in short-range axons would result in a decrease in global processing and an increase in local processing. Wang *et al.* [45], for example, using small-world network typology (characterized by dense local connections and few long connections), examined the correlation matrix between 90 cortical and subcortical regions and was able to demonstrate that networks of those diagnosed with ADHD tended toward decreased global efficiency of the brain networks over the whole cost range. Wang and colleagues [45] concluded that there is an increase in local efficiencies combined with a decreasing tendency in global efficiencies in ADHD, suggesting local (or short-range) overconnectivity and long-range underconnectivity, similar to that found in ASD [44].

It is important to note that in normal brain development, there appears to be a shift from local processing to more global processing. Uddin *et al.* [46], for example, used resting-state fMRI (rsfMRI) and revealed important principles of functional brain development, including a shift from diffuse to focal activation patterns, and simultaneous pruning of local connectivity and strengthening of long-range connectivity with age. Similarly, Fair *et al.* [47] found that development of the proposed adult control networks involves both segregation (*i.e.*, decreased short-range connections) and integration (*i.e.*, increased long-range connections) of the brain regions that comprise them. Fair *et al.* [47] stated that delay/disruption in the developmental processes of segregation and integration may play a role in disorders of control, such as ASD, ADHD, and Tourette's syndrome.

The study results are corroborated by earlier epidemiological studies that found exposure to Hg from various sources is a risk factor for HKSoC. This finding is reported in epidemiological studies regardless of the methods or the databases used. Young *et al.* [17], for example, using an ecological study design in the VSD, looked at the birth cohort prevalence of ADHD and birth cohort exposures to Hg from TCVs. They found that when infants (birth to seven months of age) were given an additional 100 µg Hg from TCVs, their risk ratio for an ADD/ADHD diagnosis increased to 3.15. In infants (birth to 13 months of age) given an additional 100 µg Hg from TCVs, their risk ratio of an ADD/ADHD diagnosis increased to 4.51.

Previously, investigators conducted a cohort study that examined the relationship between escalating exposure to Hg from TCVs and the probability of an eventual neurodevelopmental disorder diagnosis [18]. It was found that a significant dose-response relationship between increasing exposure to Hg from TCVs in infants administered within the first six months of life and the risk for an ADD/ADHD diagnosis. In still, yet another study, investigators undertook a case-control study revealing a significant dose-response relationship between exposure to Hg from TCVs administered within the first six months of life and the long-term risk of a child being diagnosed with HKSoC [48]. Finally, a recent meta-analysis of epidemiological studies examining environmental Hg exposure during embryo or early infancy and later diagnosed childhood ADHD revealed a significant association (odds ratio = 1.60, 95% confidence interval = 1.10–2.33) [49].

However, other studies have failed to show a consistent statistically significant association between HKSoC and Hg exposure from TCVs. Possibly, this is because these other studies employed different criteria for outcomes, used a childhood vaccine schedule that was dissimilar, or employed different epidemiological methods. An examination of these studies reveals that an insufficient follow-up period was utilized. Follow-up is a critically important issue in these studies that examine the relationship between exposures and the risk of diagnosed HKSoC, particularly when the exposures to all of the children in the study were similar. The reason is that the chance of a child being given an HKSoC diagnosis is not the same all through the child’s life. For instance, in this study, the initial mean age for an HKSoC diagnosis was 5.72 years old, and the SD of mean age for the initial diagnosis was 1.83 years. As such, failing to include an adequate follow-up time period that does not take into account the delay between birth and age of an initial diagnosis will not be able accurately determine the risk of diagnosed HKSoC from exposure to TCVs.

Some studies exemplify this issue. Examples are: Andrews *et al.* [50] in the General Practitioner Research Database (GPRD) and Verstraeten *et al.* [51] in the VSD. These studies used hazard models, which assume equal chances of a child being diagnosed with HKSoC with every added day of follow-up. Andrews *et al.* [50] employed Cox’s hazard ratios to determine the follow-up periods. As a result of the hazard model used, and the apparently severe lack of follow-up time for many of the members of the cohort examined, this study found that the more Hg a child was exposed to the less likely they were going to be diagnosed with ADD. As consequence, nonsensically, Hg was found to be protective for being diagnosed with ADD.

The following example from the Verstraeten *et al.* [51] study in the VSD database is particularly informative. Specifically, these investigators examined the dataset in their HMO B, a dataset that had a noteworthy overlap with the dataset that was used in the current study. Those investigators described subjects examined in their HMO B dataset were born between January 1992 through December 1998, and the dataset has follow-up data on the subjects through the end of 2000. Assuming an even distribution of births across the birth cohorts examined by Verstraeten *et al.* [51], this would suggest that the average child was born in the year 1995, and as a result, the average child would have a maximum follow-up period of five years. This is particularly troubling because the mean age of initial HKSoC diagnosis ± SD of initial diagnosis of HKSoC observed in the present study was 5.72 ± 1.83 years of age, and similarly, Verstraeten *et al.* [51] reported a median age at first diagnosis of ADD (ICD-9 code: 314.00) of 5.83 years of age. As a result, it would be reasonable to presume that there was a >50% error rate in the cohort examined of those diagnosed with ADD (*i.e.*, significantly under estimating the true number of subjects eventually to be diagnosed with ADD and overestimating the true number subjects eventually to be not diagnosed with ADD).

Furthermore, it is even possible to estimate the potential true extent of the lack of appropriate follow-up among subjects from HMO B examined in the Verstraeten *et al.* [51] study. It was reported in Verstraeten *et al.* [51] study that the risk of an ADD diagnosis was 0.90 (95% confidence interval = 0.74–1.10) for a 12.5 µg Hg exposure from TM-preserved vaccines that were given in the first month after birth. However, this finding is very different than the odds ratio (OR) of 1.45 (95% confidence interval = 1.30–1.62) that was found in the current analysis for exposure to 12.5 µg Hg from TM-HepB compared to 0 µg Hg exposure within the first month after birth in the children with an HKSoC diagnosis and controls. Examining the issue of follow-up, it should be noted that by decreasing the length of continuous enrollment among controls without a diagnosed HKSoC from the currently utilized value of 7.55 years to a much shorter time of 3.89 years (mean age of initial HKSoC diagnosis—SD of initial HKSoC diagnosis), the calculated odds ratio was 0.82 (95% confidence interval = 0.73–0.91) that was entirely compatible with that previously reported by Verstraeten *et al.* [51].

The aforementioned results are consistent with those we previously reported on subjects diagnosed with an ASD [19]. Namely, we observed that by decreasing the follow-up period among control children to only the mean age of initial diagnosis of ASD—SD of initial diagnosis of ASD, exposure to 12.5 µg Hg from TM-HepB in comparison to 0 µg Hg within the first month after birth in the children diagnosed with ASD and controls, revealed an odds ratio = 1.21 (95% confidence interval = 0.97–1.51), a value that, when taking into account the confidence intervals, was compatible with the risk ratio = 1.16 (95% confidence interval = 0.78–1.71) observed in the Verstraeten *et al.* [51] study.

### Study Limitations and Strengths

Using the VSD database, which was designed for the purpose of examining vaccine safety, is a considered strength of this study. Importantly, the study retrospectively examined prospectively collected medical records of children obtained as part of their participation in their MCOs. The children in the study had to be enrolled from birth. The cases had to be continuously enrolled until they were diagnosed with HKSoC and the control children had to be enrolled for a sufficient time period to ensure that there was only a minute possibility that they would be given a diagnosis of HKSoC during additional follow-up. As such any factors associated with enrollment were minimized. In addition, the records were examined to ensure that only those diagnosed with HKSoC after the vaccine of interest was administered were considered in the diagnosed group used in these analyses.

For those diagnosed with an HKSoC, the mean and SD of the age of the initial HKSoC diagnoses were calculated. As result, the percentage of additional diagnoses of HKSoC that might have been missed could be estimated. To ensure that most of those in the group without a diagnosis of HKSoC were truly individuals that would not subsequently be diagnosed with HKSoC, an a priori requirement was set that, to be a valid control, each control had to be enrolled in the VSD from birth continuously until their the mean age of initial diagnosis of the outcome of interest plus the SD for that mean age, which turned out to be 7.55 years of age for HKSoC. Statistically setting the follow-up age for the group without a diagnosis of HKSoC in this manner assured a less than a 16% chance that some of those in the control group might subsequently be diagnosed with HKSoC. Ideally, a longer follow-up period would have further reduced this misclassification risk. However, the limitations on the VSD patient data records that were available for review precluded a longer follow up period.

The reason it is critical to ensure that most of those in the controls are only those who have little risk of subsequently being diagnosed is that this ensures that there is minimal statistical “noise” as to the “true” diagnostic status of controls. It is very important to allow for “sufficient follow-up” of controls when conditions such as HKSoC are examined because of its long and variable-onset window. Notably, if the length of follow-up is reduced, which increases the possibility of a child in the control group being in the wrong diagnostic group, the magnitude of the observed effects could be reduced. For example, analyzing the data utilized in the present study but requiring that those in the controls to be enrolled only to 5.72 years of age (the mean age of initial HKSoC diagnosis), those in the diagnosed case group were still significantly more probable (odds ratio = 1.67, *p* < 0.01) than those in the no-diagnosis control group to have received 37.5 µg Hg from TM-HepB than those who had received 0 µg Hg within their first six months after birth. However, when the follow-up period was decreased even further, such that those in the controls had to be only enrolled only to 3.89 years of age (mean age of initial HKSoC diagnosis—SD of initial HKSoC diagnosis), those in the diagnosed cases group were statistically no more probable (odds ratio = 1.26, *p* > 0.25) than those in the controls to have received 37.5 µg Hg from TM-HepB compared to 0 µg Hg within the first six months after birth.

The fact that the data from the VSD database studied were collected on a prospective basis as part of the routine healthcare is another study strength. This is because the data were collected by healthcare providers that were completely independent of the study. As a result, potential observation and/or behavioral biases associated with certain exposures and outcomes were minimized.

Potential limitations also exist. Unknown cofounders or biases may be present in the datasets. However, outcomes that are not biologically plausibly linked to postnatal organic-Hg exposure from TM-preserved vaccines were examined, such as a diagnosis of congenital anomalies (ICD-9 code: 759.9). No significant increased risk was found when compared to a group with no congenital anomaly. Both groups were similarly exposed to 12.5 µg Hg from TM-HepB or received 0 µg Hg exposure within the first month after birth (odds ratio = 1.03, *p* > 0.50).

In addition, the present study results observed for HKSoC may possibly be due to chance. However, given the few statistical tests conducted, the highly significant findings, and the constancy of the magnitude and direction of the findings even when the data were separated by gender, this would not be likely.

It is possible that some of the children may have had more neurological problems that were not noted or the children may have been misdiagnosed. Some vaccine exposures may have been not been accurately recorded. However, these shortcomings should not have significantly affected the results. It is not certain how differential application would have occurred to affect the groups assessed based upon the TM doses that were given to the children. Misclassification would be inclined to bias the results toward the null hypothesis, because misclassification would result in the children being assigned to the wrong group and reduce the study’s statistical power.

In the present study, neurodevelopmental diagnoses other than 314.xx were not examined among cases and controls. This limitation of the present study should have had a limited impact on the results observed because of the rarity of other neurodevelopmental diagnoses as compared to a 314.xx diagnosis, but future studies could further evaluate this phenomenon.

Additionally, exposures to other sources of Hg were not determined. The children in the present study were probably exposed to other TCVs, as well as other sources of Hg, e.g., breastfeeding and infant formula. Exposure to dental amalgams, fish, or other environmental sources is also possible. However, these exposures would be inclined to bias the findings towards the null hypothesis due to confounding the specific exposure classifications of Hg examined. For example, children who were designated as having lower exposure to TCVs could have received high doses of Hg from these other sources, and children having higher exposure to TCVs could have received low doses of Hg from these other sources. This would reduce the chances of obtaining statistically significant results.

Finally, the precise timing and cumulative doses of organic-Hg from all TCVs associated with most notable consequences were not assessed. In addition, it may be useful to examine covariates such as race, weight at birth, which may further influence the extent of the effects noted.

## 5. Conclusions

This study offers further epidemiological evidence to substantiate the significant risk of increasing organic-Hg exposure from TCVs and a subsequent HKSoC diagnosis. As mentioned earlier, there are several recent studies that show that organic-Hg exposure from TCVs is a risk factor for HKSoC diagnosis. Hg is a neurotoxicant, and it has been shown to specifically target long-range axons in the brain, and as such it may be contributing to the abnormal long-range tracts found in children with an HKSoC diagnosis.

In this hypothesis-testing, epidemiological study, organic-Hg exposure from TCVs was found to be a significant risk factor for an HKSoC diagnosis. Additionally, because the children in the control group were followed for an evidenced-based follow-up period, they were appropriately assigned with respect to their exposures and outcomes. This mathematically determined follow-up period reduced the potential for confounded or biased cause-and-effect association between exposure and outcome. Future research should further examine the potential association between TCVs and other neurodevelopmental disorders, and assess the issue of timing of exposure and adverse outcomes within explicit subgroups.

Research evidence shows that Hg bio-accumulates and that Hg levels in humans and the environment are increasing [52]. In addition, Hg is able to potentiate the effects of other xenobiotics, such that combined exposure from different sources is a serious concern. Thus, all sources of Hg should be avoided, especially in vulnerable populations such as infants and children. Cumulative effects from various sources could potentially increase a child’s Hg body burden and conceivably overload their immature detoxification systems. Immunization is a way to diminish the morbidity and mortality associated with many infectious diseases [53]. However, ending the unnecessary use of TM would not reduce the effectiveness of vaccines, yet it could potentially reduce the risk of a child developing HKSoC.

## Figures and Tables

**Table 1 brainsci-06-00009-t001:** A summary of the year of birth of individuals diagnosed with hyperkinetic syndrome of childhood (HKSoC) and controls within the Vaccine Safety Datalink database.

Year of Birth	Frequency of Cases Diagnosed with HKSoC	Percent of Cases Diagnosed with HKSoC	Frequency of Controls	Percent of Controls
1991	402	23.21	6130	29.78
1992	420	24.25	9463	45.97
1993	431	24.88	4991	24.25
1994	235	13.57	-	-
1995	133	7.68	-	-
1996	63	3.64	-	-
1997	24	1.39	-	-
1998	*	*	-	-
1999	*	*	-	-
2000	*	*	-	-

*: Numbers not provided because values are less than 5; -: Absent value.

**Table 2 brainsci-06-00009-t002:** A summary of organic-mercury exposure from Thimerosal-preserved hepatitis B vaccine administration between cases diagnosed with hyperkinetic syndrome of childhood (HKSoC) and controls within the Vaccine Safety Datalink database.

Group Examined	Number Diagnosed with HKSoC (%)	Number without a Diagnosis of HKSoC (%)	Odds Ratio (95% CI)	*p*-value
Analysis I				
12.5 µg Hg exposure within 1st month	549 (36.97)	5921 (28.77)	1.45 (1.30–1.62)	<0.001
0 µg Hg exposure within 1st month	936 (63.03)	14,663 (71.23)		
Analysis II				
25 µg Hg exposure within first 2 months	550 (37.39)	5930 (29.48)	1.43 (1.28–1.59)	<0.001
0 µg Hg exposure within first 2 months	921 (62.61)	14,185 (70.52)		
Analysis III				
37.5 µg Hg exposure within first 6 months	112 (76.71)	725 (42.18)	4.51 (3.04–6.71)	<0.001
0 µg Hg exposure within first 6 months	34 (23.29)	994 (57.82)		

**Table 3 brainsci-06-00009-t003:** A summary of organic-mercury exposure from Thimerosal-preserved hepatitis B vaccine administration between male cases diagnosed with hyperkinetic syndrome of childhood (HKSoC) and male controls within the Vaccine Safety Datalink database.

Group Examined	Number Diagnosed with HKSoC (%)	Number without a Diagnosis of HKSoC (%)	Odds Ratio (95% CI)	*p*-value
Analysis IV				
12.5 µg Hg exposure within 1st month	424 (36.61)	2943 (28.56)	1.44 (1.27–1.64)	<0.001
0 µg Hg exposure within 1st month	734 (63.39)	7360 (71.44)		
Analysis V				
25 µg Hg exposure within first 2 months	425 (36.99)	2947 (29.31)	1.42 (1.25–1.61)	<0.001
0 µg Hg exposure within first 2 months	724 (63.01)	7109 (70.69)		
Analysis VI				
37.5 µg Hg exposure within first 6 months	90 (76.92)	353 (42.22)	4.56 (2.90–7.16)	<0.001
0 µg Hg exposure within first 6 months	27 (23.08)	483 (57.78)		

**Table 4 brainsci-06-00009-t004:** A summary of organic-mercury exposure from Thimerosal-preserved hepatitis B vaccine administration between female cases diagnosed with hyperkinetic syndrome of childhood (HKSoC) and female controls within the Vaccine Safety Datalink database.

Group Examined	Number Diagnosed with HKSoC (%)	Number without a Diagnosis of HKSoC (%)	Odds Ratio (95% CI)	*p*-value
Analysis VII				
12.5 µg Hg exposure within 1st month	125 (38.23)	2978 (28.97)	1.52 (1.21–1.90)	<0.001
0 µg Hg exposure within 1st month	202 (61.77)	7303 (71.03)		
Analysis VIII				
25 µg Hg exposure within first 2 months	125 (38.82)	2983 (29.66)	1.51 (1.2–1.89)	<0.001
0 µg Hg exposure within first 2 months	197 (61.18)	7076 (70.34)		
Analysis IX				
37.5 µg Hg exposure within first 6 months	22 (75.86)	372 (42.13)	4.32 (1.83–10.21)	<0.001
0 µg Hg exposure within first 6 months	7 (24.14)	511 (57.87)		

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
