# Peer review of "Thimerosal-Preserved Hepatitis B Vaccine and Hyperkinetic Syndrome of Childhood"

_brainsci, 2016, doi:10.3390/brainsci6010009_

Round 1

Reviewer 1 Report

I highly recommend this manuscript for publication with minor revisions. This is a straightforward analysis of an important database with profound implications for public health. The Odds ratios presented are dose dependent and, above 37.5 uG of Hg, they reach over 4.5 which are tremendous associations between an exposure and a disease outcome. These results are relevant and important as thimerosal is still included in global vaccines although they have largely been removed from childhood vaccine in the United States. The writing is clear, the analysis is sound, and the presentation is excellent with a few exceptions.

1. In table 1, the "-" is used to signify absent values but that should be footnoted in the table precisely what this signifies. 

2. In the exposed and control groups were there other related diseases such as Autism? If so, how many and what would happen to the analysis if you removed them?

3. On page 3 line 22, they describe children with the diagnosis before the exposure. How many were there that fit in this category?

4. On page 11 line 4 you say levels of Hg in the environment and in humans are increasing but your reference #26 doesn't speak to that point. Are there better references to substantiate this point?

5. Can the authors address this point: is this database publicly available?

Author Response

Comment 1. In table 1, the "-" is used to signify absent values but that should be footnoted in the table precisely what this signifies. 

Response to comment: In the revised manuscript, this was corrected as suggested.

Comment 2. In the exposed and control groups were there other related diseases such as Autism? If so, how many and what would happen to the analysis if you removed them?

Response to comment: Both cases and controls could have been diagnosed with autism, but it should have been a relative minority. Autism is relatively rare compared to a 314 diagnosis. This issue was added to the study limitations in the revised manuscript.

“In the present study, neurodevelopmental diagnoses other than 314.xx were not examined among cases and controls. This limitation of the present study should have had a limited impact on the results observed because of the rarity of other neurodevelopmental diagnoses as compared to a 314.xx diagnosis, but future studies could further evaluate this phenomenon.”

Comment 3. On page 3 line 22, they describe children with the diagnosis before the exposure. How many were there that fit in this category?

Response to comment: We don’t have the exact number, but it is assumed to be a relatively small number.

Comment 4. On page 11 line 4 you say levels of Hg in the environment and in humans are increasing but your reference #26 doesn't speak to that point. Are there better references to substantiate this point?

Response to comment: In the revised manuscript, this was corrected with a better reference.

Comment 5. Can the authors address this point: is this database publicly available?

Response to comment: In the revised manuscript, this information was added.

“This database is available to outside researchers after obtaining approval from the CDC and KP. Information regarding access to this database is at: http://www.cdc.gov/vaccinesafety/ensuringsafety/monitoring/vsd/accessing-data.html.”

Reviewer 2 Report

This is an important study that deserves to be published.  The results are controversial. However, the findings may be of great interest for the readers of the journal. The paper is of high quality. I recommend the manuscript published. It can be published as it is.

Author Response

Thank you.